# SARS-CoV-2 variant survey: Comparison of RT-PCR screening with TGS and variant distribution across two divisions of Bangladesh

**Zannat Kawser**[1][☉]*, **Saikt Rahman**[1][☉], **Emilie Westeel**[2], **Mohammad Tanbir Habib**[1], **Mohabbat Hossain**[1,3], **Md. Rakibul Hassan Bulbul**[1,4], **Sharmin Aktar Mukta**[1], **Md. Zahirul Islam**[1,4], **Md. Zakir Hossain**[4], **Mokibul Hassan Afrad**[5], **Manjur Hossain Khan**[6], **Tahmina Shirin**[6], **Md. Shakeel Ahmed**[4‡], **Jean-Luc Berland**[2], **Florence Komurian-Pradel**[2‡], **Firdausi Qadri**[1,5‡]

1 Institute for Developing Science and Health Initiatives, Dhaka, Bangladesh, 2 Fondation Mérieux, Direction Médicale et Scientifique, Lyon, France, 3 Department of Genetic Engineering and Biotechnology, University of Chittagong, Chittagong, Bangladesh, 4 Bangladesh Institute of Tropical and Infectious Disease (BITID), Fouzderhat, Chittagong, Bangladesh, 5 International Centre for Diarrheal Disease Research, Bangladesh, Dhaka, Bangladesh, 6 Institute of Epidemiology, Disease Control, and Research, Dhaka, Bangladesh

☉ These authors contributed equally to this work.
‡ MSA, FKP and FQ are senior authors on this work.
* zannatzkawser@gmail.com

**Data Availability Statement:** All relevant data are available in the paper and supporting information files.

## Abstract

### Background

The widespread increase in multiple variants of severe acute respiratory syndrome corona-virus-2 (SARS-CoV-2) since 2020 is causing significant health concerns worldwide. While whole-genome sequencing (WGS) has played a leading role in surveillance programs, many local laboratories lack the expertise and resources. Thus, we aimed to investigate the circulating SARS-CoV-2 variants and evaluate the performance of multiplexed real-time reverse transcription-PCR (RT-PCR) for screening and monitoring the emergence of new SARS-CoV-2 variants in Bangladesh.

### Methods

A total of 600 confirmed SARS-CoV-2-positive cases were enrolled either prospectively or retrospectively from two divisions of Bangladesh. The samples were screened by variant RT-PCR targeting five mutations of the spike gene (N501Y, P681R, L452R, E484K, E484Q). A subsample of the study population was also selected for third-generation sequencing (TGS) and the results were compared to the variant RT-PCR screening. An in-depth comparison was made between the two methods in terms of congruence and cost-benefit.

**Funding:** The authors received no specific fundingfor this work.

**Competing interests:** The authors declare no competing interest.

## Result

Seven variants were detected among samples, with similar distributions of the variants across both divisions. Variant RT-PCR for the targeted mutations lead to a 98.5% call rate; only nine samples failed to be determined. No association was found regarding the demographic features, clinical criteria, or routine RT-PCR Ct values across the variants. The clade diversity of the sequenced subpopulation ($n = 99$) exhibited similar distributions across the two study sites and other epidemiologic variables. Variant RT-PCR successfully distinguished variants of concern (VOCs) and variants of interest (VOIs); however, 8% discrepancy was observed for the closest lineages. Moreover, the variant RT-PCR represented an ideal balance of cost, time, and accuracy that outweigh their limitations.

## Conclusion

Based on the strong agreement of variant RT-PCR with TGS, such rapid, easily accessible approaches of rapid strain typing are essential in the context of pandemic responses to guide both treatment decisions and public health measures.

## Introduction

As COVID-19 spreads, the novel Severe Acute Respiratory Syndrome Coronavirus 2 (SARS-CoV-2) naturally mutates to form new variants that can either be more or less transmissible and infectious than the wildtype form, depending on the alterations [1]. Similarly to the global pandemic, Bangladesh has also experienced waves of COVID-19 due to the emergence of SARS CoV-2 variants of concern (VOCs), including the Alpha (B.1.1.7) (201/501Y.V1I), Beta (B.1.351) (20H/501Y.V2), Delta (B.1.617.2), and Omicron (B.1.1.529.1 & B.1.1.529.2) variants. The emergence of the Kappa (B.1.617.1) variant just preceded the first surges of Delta and Zeta (P.2); these variants were detected in Bangladesh and globally and were designated as variants of interest (VOI) from an epidemiological point of view [2, 3].

Since its emergence, many molecular variants of SARS-CoV-2 with differences in the spike (S) glycoprotein have been detected; these variants enable the virus to infect the host more effectively [4]. Most mutations of interest occur in this S protein, the viral protein that initiates the attachment of the virus to the host cell receptor. Thus, these variants alter the efficiency of antibody protection [5, 6]. Since late 2020, several biologically significant S mutations have been associated with increased virulence, transmission, pathogenicity, and antibody escape [7]. A combination of mutations (L452R, E484(K/Q), N501Y) at three positions within the spike protein result in the Alpha, Beta, Gamma, Delta, and Kappa variants, which have spread globally [8–10]. The Alpha variant has an N501Y mutation at the 501 residue, resulting in replacement of asparagine (N) with tyrosine (Y), which confers consistent fitness gains for replication in human airway cells [11]. The Beta variant carries an E484K mutation, glutamic acid (E) replaced with lysine (K), along with the N501Y mutation. Delta and Kappa share two common mutations, P681R and L452R (leucine [L] replaced by arginine [R]) [12, 13]. The P681R substitution enhances viral replication as a result of improved transmission and replication capacity, and is likely to have been the main driver of the global Delta surge [14]. In addition to these two mutations, Kappa harbors E484Q (glutamic acid [E] substituted by glutamine [Q]) [12]; E484Q was also reported to be present in some strains of Delta isolated in Southeast Asia [8] and other countries [15]. Omicron replaced Delta, and contains the clade-defining

mutation (CDM) N501Y, which is not present in the Delta variant [16]. Surveillance for these mutations has been embedded in the surveillance programs initiated by national health authorities. However, more cost-effective strategies targeting these mutations are needed to monitor for shifts in the distribution of SARS-CoV-2 variants.

The gold standard method for tracking emerging variants, whole genome sequencing (WGS), is costly, takes considerable time to complete, and requires expert analysis, which can limit its use in local settings [17]. In contrast, variant detection using RT-PCR is a feasible, simpler, less expensive, and faster technique that can be quickly adapted to detect newly identified VOC mutations that affect the spike protein [9]. Real-time identification of VOCs and VOIs can have a significant beneficial impact on outbreak responses [18, 19] and patient care.

The continuous emergence of VOCs highlights that enhanced ongoing surveillance is needed to investigate the phylodynamics of SARS-CoV-2 variants and their related clinical features. An accurate, low-cost RT-PCR variant screening strategy would enable more selective use of WGS by targeting samples of interest. This would maximize the use of WGS resources and broaden the ability of local clinical laboratories to actively participate in identification of VOCs. Thus, in this study, we aimed to compare the efficacy, congruence, and cost-benefit of real-time PCR and TGS screening methods and conduct a preliminary sentinel surveillance of changes in variant composition and the structure of SARS-CoV-2 at the local laboratory level in two districts of Bangladesh.

## Materials and methods

### Study population and data collection

A total of 600 nasopharyngeal swab (NPS) specimens in viral transport media (VTM) were obtained in the Dhaka (n = 300) and Chattogram (n = 300) divisions of Bangladesh. The samples from Dhaka division were collected over 21 months, from November 2020 to July 2022, and the samples from Chattogram, from April 2020 to May 2022. Then the samples were tested for routine diagnostic RT-PCR assay using commercially available diagnostic kits targeting ORF1ab and N gene. The NPS specimens with Ct value <30 were randomly selected as per the TaqMan™ SARS-CoV-2 Mutation Panel user guide for undergoing Variant RT-PCR. Symptomatic and asymptomatic cases representing all age groups were enrolled in the study both prospectively and retrospectively. Approval from the Institutional Review Board of the Institute of Epidemiology, Disease Control and Research (IEDCR) (# IEDCR/IRB/19; Date of approval: 09 November, 2021) was obtained for the study. All procedures were performed in compliance with the ethical guidelines. Since there was no direct contact with the study participants and data was retracted throughout the mentioned study duration from the Data Collection Form approved by the Government of Bangladesh for routine diagnostic use, separate informed written consent was not obtained for the present study. After taking oral consent via phone, the health outcome data was obtained. All accessible information from individual participants was documented and recorded using the Castor Electronic Data Capture (EDC) tool.

### Viral RNA extraction

Viral RNA was extracted from the NPS manually using QIAamp Viral RNA Mini Kits (Qiagen, Germany) on an automated NEXOR-32 extraction system (China) according to the user manual. Then, viral RNA samples were used to perform variant RT-PCR screening. The remainder of the VTM containing the NPS was stored at -70˚C.

## SARS-CoV-2 variant RT-PCR screening

Specimens that were positive for SARS-CoV-2 by routine diagnostic RT-PCR were examined for spike-gene variants via a combination of allele-specific primers and allele-specific detection technology using a TaqMan™ SARS-CoV-2 Mutation Panel (Applied Biosystems by Thermo Fisher Scientific, U.S.A) that targets five mutations (N501Y, P681R, L452R, E484K, E484Q) present in the S-gene of SARS-CoV-2.Briefly, 5 μL total nucleic acid eluate was added to a 15 μL total-volume reaction mixture (5 μL TaqPath™ 1-Step RT-qPCR Master Mix, CG [4X], 0.5 μL TaqMan™ SARS-CoV-2 Mutation Panel Assay [40X], and 9.5 μL nuclease-free water, with 0.9 mM of each primer and 0.2 mM of each probe). The variant RT-PCR was performed on a CFX96 (C1000 Touch) Real Time System (Bio-Rad, USA). The S-N501Y, S-E484K/Q, S-L452R, and S-P681R assays were carried out following the running conditions specified in the procedure manual (TaqMan™ SARS-CoV-2 Mutation Panel User Guide, Publication Number MAN0024768). One VIC dye labeled probe was used to detect the reference sequence and one FAM dye labeled probe was used to detect the mutant sequences, from each of the wells in PCR plates containing one of the five mutation sequences. Variant RT-PCR data was analyzed using QuantStudio™ Design and Analysis Software v2.5.

The variants were determined based on the amino acid calls at the four targeted sites, as follows: wildtype: no mutations detected at any of the four positions; probable Delta: P681R, L452R, and no mutations at the other two sites; probable Omicron: N501Y and no mutations at the other three sites, or N501Y, L452R, and no mutations at the other two sites (in samples with dates of collection after the December 2021 Delta surge); probable Kappa: P681R, L452R, E484Q, and no mutations at the other site; probable Beta: N501Y, E484K, and no mutations at the other two sites; probable Alpha: N501Y and no mutations at the other three sites (in samples with dates of collection before the June 2021 Delta outbreak); probable Zeta: E484K and no mutations at the other three sites; undetermined: P681R and no mutations at the other three sites (Table 1).

## Targeted third-generation sequencing using the MinION platform

The set of samples for sequencing were selected from the 600 study samples based on the following criteria: 1) samples with low Ct (<27) values, 2) the very first sample for which variant RT-PCR called a new variant in the study timeframe, 3) the last sample for which variant RT-PCR called a variant in the study timeframe, and 4) the most recently collected samples (from May, June, and July 2022); these criteria were applied to the samples collected from both divisions of Bangladesh.Based on these criteria, a total of 100 variant PCR-screened SARS-CoV-2 positive specimens (50 from Dhaka division and 50 from Chattogram division) were sequenced on the MinION platform.

**Table 1. Interpretation of variant RT-PCR based on five mutations.**

| Probable Variant | N501Y | L452R | P681R | E484K | E484Q |
|---|---|---|---|---|---|
| Wild type | No call | No call | No call | No call | No call |
| Probable Alpha | Mutation | No call | No call | No call | No call |
| Undetermined | No call | No call | Mutation | No call | No call |
| Probable Beta | Mutation | No call | No call | Mutation | No call |
| Probable Zeta | No call | No call | No call | Mutation | No call |
| Probable Delta | No call | Mutation | Mutation | No call | No call |
| Probable Kappa | No call | Mutation | Mutation | No call | Mutation |
| Probable Omicron | Mutation | No call | No call | No call | No call |
| Probable Omicron | Mutation | Mutation | No call | No call | No call |

For MinION sequencing via Oxford Nanopore Technology (ONT), a cDNA library was generated for each sample using LunaScript RT SuperMix Kit (New England Biolabs, USA) utilizing either ARTIC v3 or ARTIC v4 primer-based multiplex PCR to generate amplicons with Q5 high-fidelity DNA polymerase (New England Biolabs). dA-tailing of the amplified amplicons was performed utilizing the NEBNext Ultra II end repair/dA-tailing module (New England Biolabs), followed by ligation to the native barcodes EXP-NBD104 and EXP-NBD114 (Oxford Nanopore Technologies) using blunt/TA DNA ligase (New England Biolabs). The resultant sequencing libraries were pooled, ligated with an adapter, and then sequenced via FLO-MIN106D flow cells for at least six hours. MinKNOW v21.02.1 (bionic) was used to accomplish base calling and demultiplex the raw reads. Processed reads were assembled using the ARTIC gupplyplex code script with Medaka v1.4 on the EPI2ME desktop agent v3.5.7 (FastQ quality control plus ARTIC plus NextClade) (https://artic.network/ncov-2019/ncov2019-bioinformatics-sop.html). The consensus genome was generated for each library and the aligned file was visualized using AliView, followed by manual verification and correction of any premature stop codons and/or frame shifts. One sample was excluded from further analysis due to low coverage.The FASTA sequences have been uploaded to the GenBank public sequence database (see S1 Table) and the FASTQ files have been submitted to the SRA database. Both FASTA and FASTQ files are publicly accessible from December, 2023.

## Variant detection and phylogenetic analyses

Both NextClade (https://clades.nextstrain.org) and Pangolin (https://github.com/cov-lineages/pangolin) were used for variant and lineage detection (using the NextClade database downloaded on April 25th, 2023). The phylogenetic tree was computed using the IQ-TREE 2.0.3. Consensus sequences were aligned to the SARS-CoV-2 reference sequence (NC_045512.2) using NextClade 2.11.0 and the alignment was passed to IQ-TREE with automatic model selection and 1000 bootstrap values, then the constructed phylogenetic tree was visualized using the R application ggtree.

## Accuracy, precision, and data analyses

To assess the accuracy of variant screening, in addition to assessing the five mutations at four positions in the S-gene of SARS-CoV-2 targeted by RT-PCR, we also considered all other clade-defining mutations (CDMs) within the genome to compare the outcomes of the variant RT-PCR and TGS analyses (https://covariants.org/). Statistical evaluations of epidemiological and clinical data were performed using Fisher's exact test, the Chi-squared test, or the Wilcoxon-Mann-Whitney test.

# Results

## Baseline characteristics of the study population

A total of 600 SARS-CoV-2 positive cases were enrolled in the study: 313 samples were collected prospectively and 287 were collected retrospectively, with the samples equally distributed across two major divisions of Bangladesh, Dhaka and Chattogram. The basic and demographic characteristics of the patients from whom the samples were collected are shown in Table 2. Most cases were symptomatic and had classic COVID-19 symptoms, including fever, cough, loss of taste and smell, or headache. Cardiovascular disease including hypertension and diabetes were the most common comorbidities.

The vaccinated individuals in this study had received either one or two types of five vaccines, namely VAXZEVRIA-COVID-19 Vaccine AstraZeneca, COMIRNATY Pfizer-

**Table 2. Basic and clinical characteristics of the patients with COVID-19 (*N* = 600).**

| Characteristic | Number (%) |
|---|---|
| **Age** (years) | |
| Range | 2 to 87 |
| Median | 40 |
| **Sex** | |
| Male | 367 (61%) |
| Female | 233 (39%) |
| **Vaccination status** | |
| Not vaccinated | 109 (18%) |
| Received one dose | 29 (5%) |
| Received two doses | 313 (52%) |
| Received more than two doses | 149 (25%) |
| **Clinical features** | |
| Symptomatic | 515 (86%) |
| Fever | 427 (83%) |
| Cough | 332 (64.4%) |
| Sore throat | 78 (15%) |
| Loss of smell | 74 (14.4%) |
| Loss of taste | 65 (12.6%) |
| Headache | 67 (13%) |
| Others* | 94 (18%) |
| Asymptomatic | 85 (14%) |
| **Underlying conditions** | |
| Present | 130 (21.8%) |
| Cardiovascular disease (including hypertension) | 76 (58%) |
| Diabetes | 72 (55%) |
| Asthma | 5 (3.8%) |
| Chronic liver or kidney disease | 4 (3%) |
| Others | 6 (4.5%) |
| Absent | 439 (73.2%) |
| Unknown | 31 (5%) |
| **Health outcome** | |
| Recovered | 588 (98%) |
| Died | 5 (0.8%) |
| Other (not yet recovered, unknown) | 7 (1.2%) |

*Difficulty in breathing, muscle ache, generalized body ache, joint pain, chest pain, common cold and runny nose

BioNTech, SPIKEVAX-COVID-19 Vaccine Moderna, SINOPHARM-WUHAN COVID-19 Vaccine, and COVID-19 Vaccine Janssen (Johnson & Johnson). The majority of patients had received two doses of vaccine (52%), followed by patients who had received more than two doses (25%) and those who had not received any vaccination at all (18%).

The distribution of routine RT-PCR Ct values across different age groups (S1 Fig), and across symptomatic and asymptomatic cases (S2 Fig) is plotted. Most of our study population were in the 20 to 40 (45%) or ≥40 to 60 (36%) age groups. The most frequent Ct value for the study specimens tested with the variant PCR screening was 22.5±4.08. There was a trend towards lower mean routine RT-PCR Ct values in the <20 age group compared to all other

age groups. However, there were no significant differences between the median Ct values of the asymptomatic and symptomatic groups or between the different age groups ($P > 0.05$).

## Variants detected by RT-PCR

A total of seven variants were detected by variant RT-PCR among the 600 NPS, with a similar distribution of variants across both divisions included in the study. The interpretation of the RT-PCR screening for variant detection was based on allelic signals at targeted positions, as well as the date of sample collection. RT-PCR identified 211 samples with the N501Y mutation after the Delta surge (probable Omicron); 30 samples with the same mutation before the Delta outbreak (probable Alpha); 279 samples with both the P681R and L452R mutations (probable Delta); 35 samples with both the N501Y and E484K mutations (probable Beta); eight samples with the P681R, L452R, and E484Q mutations (probable Kappa), and two samples with only the E484K mutation (probable Zeta). None of the five mutations were detected in 26 samples, and these samples were assumed to be the wildtype virus. The variant could not be interpreted for nine samples with mutations only observed at the 681 position.

S3A Fig presents the distribution of the Ct values across different variants detected by routine RT-PCR in the samples collected from April 2020 to July 2022. The lowest Ct values, corresponding to high viral loads, were observed during June to August 2021 during the second surge of Delta and during February 2022, when Omicron was the sole circulating SARS-CoV-2 variant in the country. The significant differences among the mean Ct values for each variant are shown in S3A Fig. The Ct values of the five primer probes in the 600 study samples, were congruent with the Ct values of the routine SARS-CoV-2 RT-PCR as plotted in S3B Fig.

The wildtype and undetermined variants were only detected among individuals who exhibited symptoms during the early months of the pandemic. In contrast, the Delta and Omicron variants were observed among people both with and without COVID-19 symptoms (symptomatic 89% vs. asymptomatic 11% for Delta; symptomatic 85% vs. asymptomatic 15% for Omicron), as shown in Table 3.

The profiles and patterns of emergence of the eight SARS-CoV-2 variants detected by variant RT-PCR over the study period in both Dhaka and Chattogram divisions of Bangladesh are presented in Fig 1. The wildtype dominated from April 2020 until July 2022, followed by emergence of the Alpha, Beta, Delta, and Omicron variants. A small number of Kappa variants were detected by variant RT-PCR screening during the period of emergence of the Delta

**Table 3. Associations between the variants detected by RT-PCR and the presence or absence of COVID-19 symptoms.**

| Variants by RT-PCR (N) | Symptomatic, N (%) | Asymptomatic, N (%) | **P-value |
|---|---|---|---|
| Alpha (30) | 28 (93.33%) | 2 (6.67%) | 0.014 |
| Beta (35) | 24 (68.57%) | 11 (31.43%) | 0.0001 |
| Delta (278)* | 247 (88.85%) | 31 (11.15%) | 0.0003 |
| Kappa (8) | 8 (100%) | 0 (0%) | - |
| Omicron (211) | 170 (80.57%) | 41 (19.43%) | 0.0001 |
| Undetermined (9) | 9 (100%) | 0 (0%) | - |
| Wildtype (26) | 26 (100%) | 0 (0%) | - |
| Zeta (2) | 2 (100%) | 0 (0%) | - |

*For one Delta variant, information on symptoms was unknown.

**Pearson's Chi-squared test with simulated p-value (based on 10000 replicates).

P-values could not be determined for the same proportions of symptomatic and asymptomatic cases infected with the Kappa, Zeta, and undetermined variants and the wildtype virus.

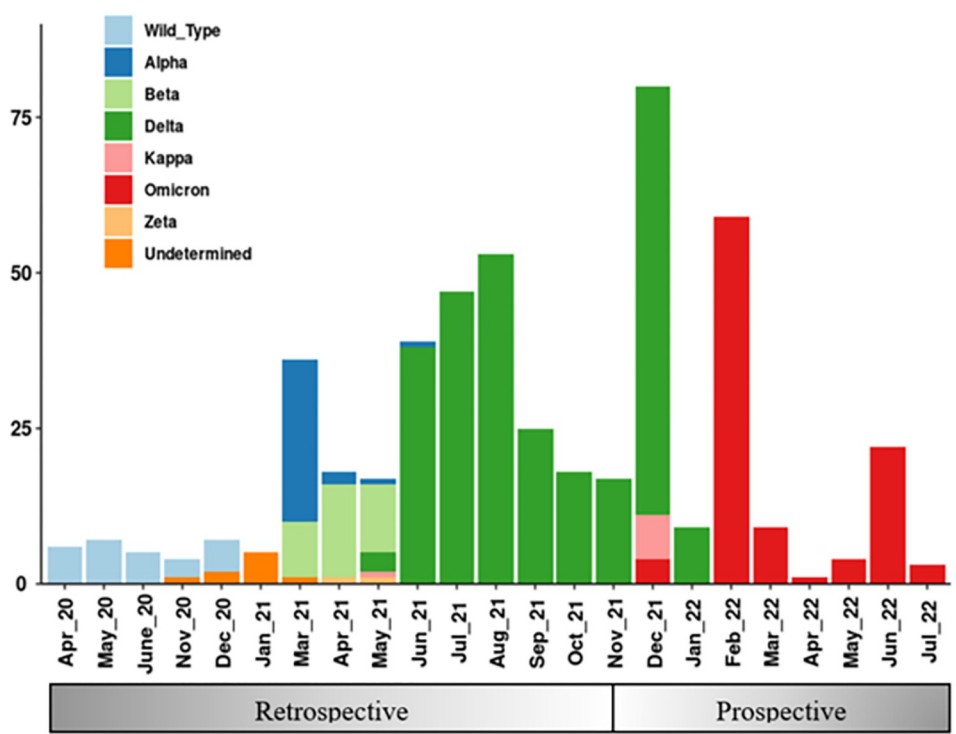

**Fig 1. Distribution and patterns of emergence of the probable SARS-CoV-2 variants detected by RT-PCR over the study time frame.**

variant in May 2021 and replacement of Delta by Omicron in December 2021.There was no specific association between the timing of sample collection for the nine samples collected between November 2020 and March 2021 that could not be categorized as any particular variant using the variant RT-PCR mutation panel. The frequency distributions of the probable variants detected by variant RT-PCR between April 2020 to July 2022 are presented in S2 Table.

## Phylogenetic analysis

The results of TGS were compared to the results of the variant RT-PCR screen for a subset of 99 specimens. Overall, 92% of the variant RT-PCR calls agreed with the results of TGS on the MinION platform (Fig 2), including 100% of the variant RT-PCR calls of the Omicron variant. The raw reads of all 99 samples have been submitted to GenBank under the Bioproject PRJNA1049588. The lab sample ID and corresponding GenBank accession IDs are given in S2 Table.

In this study, discordant results were observed for nine samples between the two methods (Table 4); of these nine samples, eight were VOIs or VOCs.

One sample identified as a probable Alpha variant by variant RT-PCR was called 20B by NextClade and had a 681H signal that was neither a target allele nor a CDM for 20B and a basecalling error at position 484. Sample BD_B_210707_653 was called 'probable Delta' by variant RT-PCR based on the presence of mutant alleles, while the NextClade analysis called this isolate 20B due to a basecalling error at position 452. Four samples were determined as Kappa by variant RT-PCR (based on the presence of L452R, P681R, and E484Q), whereas NextClade analysis identified these samples as Delta due to the presence of 32/37 other CDMs across the SARS-CoV-2 genome. Two samples were identified as Zeta by variant PCR due to a

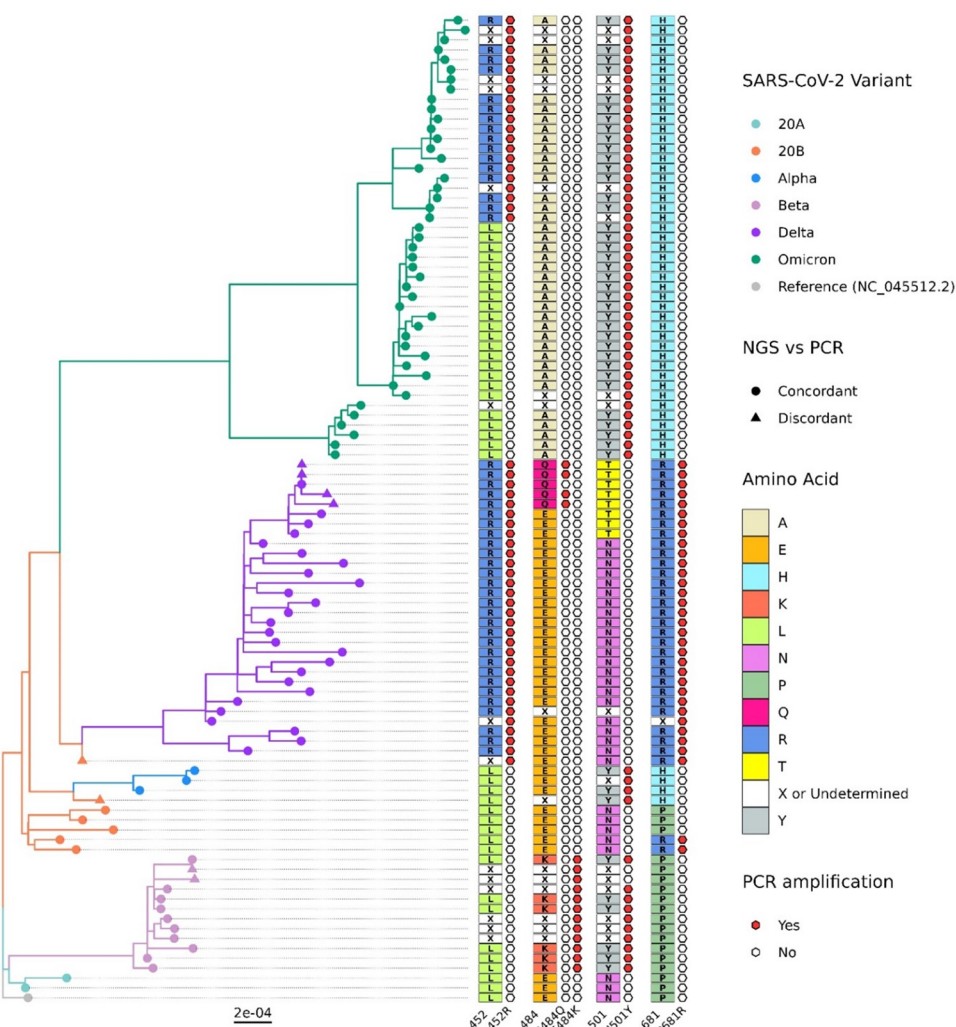

**Fig 2. Phylogenetic analysis and comparison of the outputs of variant RT-PCR and TGS for the subsample of SARS-CoV-2 isolates (*n* = 99).** The phylogenetic tree displays the 99 samples sequenced by Oxford Nanopore Technologies. Branches and nodes are color-coded based on the variants identified by NextClade (left). Inferred amino acids from TGS at the four specific positions targeted by variant RT-PCR on the spike gene of SARS-CoV-2 are highlighted by the type of amino acid (center). The hexagons next to each ONT-identified amino acid column indicate the amino acid calls based on variant RT-PCR signals. The corresponding hexagons are colored red if signals were captured for mutant alleles at the respective position.

**Table 4. Discrepancies between variant PCR and NextClade interpretation after TGS.**

| Sample-ID | Variant PCR result | NextClade result |
|---|---|---|
| BD_B_210311_183 | Alpha | 20B |
| BD_B_210707_653 | Delta | 20B |
| BD_B_211228_12 | Kappa | Delta |
| BD_IVPP_145 | Kappa | Delta |
| BD_IVPP_147 | Kappa | Delta |
| BD_B_211212_05 | Kappa | Delta |
| BD_IVPP_177 | Zeta | Beta |
| BD_IVPP_208 | Zeta | Beta |

484K mutation; however, these samples were classified as Beta by NextClade due to the presence of 20/26 CDMs for BD_IVPP_177 and 18/26 CDMs for BD_IVPP_208 assigned to Beta.

Finally, we compared the turnaround time (TAT) and cost effectiveness of both methods. In this study, we used 96-well-plates for the variant RT-PCR screen, in which 12 specimens could be tested with five primer-probe sets per thermal cycler run, and used a FLO-MIN106 flowcell to sequence 24 samples per run. For variant RT-PCR, the TAT is three hours and the approximate cost (including human resources, cost of nucleic acid extraction and reagents, and other indirect expenses) is $40 per sample. For TGS by ONT, the TAT is three days and the cost per sample is $100. The MinION system allows 12/24/96 specimens per run, while the number of specimens per run for variant RT-PCR method depends on the number of primers targeted and type of PCR plate used (96/384-well-plate).

## Discussion

The surveillance of SARS-CoV-2 variants has primarily been based on WGS, the gold standard approach for investigating the genetic makeup of pathogens. However, the availability of WGS is insufficient in many laboratories because of limited Next Generation Sequencing (NGS) and TGS tools. Continuous monitoring and screening for the emergence of SARS-CoV-2 variants necessitates the use of other accurate, simpler, and more cost-effective and time-efficient methods to guide infection control and treatment decisions in a timely manner. This study further confirms that rapid variant typing by RT-PCR can enhance surveillance efforts to deal with this critical demand and demonstrates that variant RT-PCR can be used to accurately and cost-effectively monitor the emergence of new SARS-CoV-2 variants in Bangladesh compared to T.

SARS-CoV-2 has consistently mutated throughout the course of the pandemic, resulting in a number of variants. Over the wide time frame of this study, sampling could not be performed proportionately due to variations in the frequency of viral detection of SARS-CoV-2 in the country. As a result, the distribution of the total number of cases was not proportional in terms of the different variants detected. However, in accordance with the circulating variants in Bangladesh, the patterns of the variants detected by variant RT-PCR over the time frame of this study were similar to previous reports [2, 5, 20]. The Thermo Taqman SARS-CoV2 TaqMan SARS-CoV-2 Mutation Panel was one of the first released test. As it is not a kit, one can switch in between primers/probes' sets to adapt to the changing major variants along the time frame. The chosen set was enough to capture the variants along the study time frame.

The routine RT-PCR Ct values or viral loads of the specimens collected from both Dhaka and Chattogram are similar to those of samples from studies of COVID-19 patterns in other geographic areas of Bangladesh, as well as neighboring countries [21, 22]. In addition, no significant difference in the median Ct values was observed between symptomatic and asymptomatic cases in this study, similar to previous reports [23, 24]. Notably, since follow-up specimens were available, some of these cases may have been pre-symptomatic, rather than asymptomatic. We observed a trend towards lower Ct values during the Delta and Omicron peaks in the country. However, we cannot infer any link between the routine RT-PCR Ct values and underlying epidemiologic changes at the population level or the variant profiles. Factors including vaccination status, duration from symptom onset to testing, and the use of different kits at study sites may have considerable impacts on the associations between the sample routine RT-PCR Ct values and these parameters.

The variant RT-PCR assay employed in the present study assesses the most commonly occurring mutation sites in the spike protein that affect neutralization of the SARS-CoV-2 virus by either monoclonal antibodies or convalescent plasma [25]. The non-synonymous

CDMs described in Covariants (https://covariants.org/) were also considered in our TGS analysis. The variant RT-PCR assay in our study targeted five mutations (N501Y, L452R, E484K, E484Q and P681R) at four positions. This mutation panel was selected based on the signature characteristics of the amino acid substitutions in the spike gene described in NextClade, as these changes are responsible for the emergence of new variants. The variant RT-PCR successfully detected VOIs and VOCs (as represented in the phylogenetic tree) and was congruent with the TGS interpretation in 92% of samples. In-depth comparative analysis was conducted to identify the causes of the discrepancies between the two methods. We only found discordant results for closely relatedvariants during the emergence and shifts of new variants. Detection of the E484Q mutation was only reported in the Kappa variant in earlier studies [26–28]. This mutation was later acquired by the Delta variant [29], and according to NextClade, has become another non-synonymous CDM of Delta. One out of the five amino acid calls for 484Q detected by TGS was missed by the variant RT-PCR and this sample was therefore misinterpreted as Delta instead of Kappa; however, the RT-PCR calls for the other four samples with this mutation were concordant with the TGS. However, the variant RT-PCR primer set designed to distinguish the Kappa and Delta variants based on signals from E484Q was incongruent with variant calls by TGS and was later discontinued by the supplier. In the other discrepant cases, the ONT-generated data was not conclusive due to basecalling errors at different positions, including the targeted signature sites.

## Limitations

The range of variants that can be detected by variant RT-PCR is limited to the primers designed for known variants. In this study, five mutations at four positions of the S gene were targeted. Including more targets in the panel would increase the likelihood of defining the undetermined, Kappa, and Zeta variants. While not tested in this study, variant RT-PCR may not be able to detect for hybrids or recombinants such as XBB and BQ.1. Another limitation is the low coverage of the mutation sites leading to the multiple base calling errors in ONT. As a result, a few discordances between the variant RT-PCR vs. TGS analysis could not be concluded on only the basis of the S gene CDMs using NextClade.

## Conclusions

The current study suggests that if the dominance of the primary circulating strain is high enough in a population, the need for reconfirmation of variant PCR screens by sequencing may become minimal. In this context, variant screening by RT-PCR could be used to facilitate rapid and accurate identification and inform health policy makers to take necessary measures. However, it will be necessary to update the primers used in the mutation panel as new variants emerge, in order to capture the rapid viral evolution of SARS-CoV-2. The larger the number of mutations targeted by this screening method, the more accurate the detection of variants.

## Supporting information

**S1 Table. Specimen ID and corresponding GenBank accession ID for 99 sequences.** (DOCX)

**S2 Table. Probable SARS-CoV-2 variants detected by variant RT-PCR from April 2020 to July 2022 in Dhaka and Chattogram, color-coded as frequency distributions.** (DOCX)

**S1 Fig. Distribution of the Ct values of the NPS specimens by age group.** No significant difference of median Ct values was noted in different age groups.
(DOCX)

**S2 Fig. Distribution of the Ct values of the NPS specimens in cases with and without symptoms of COVID-19.** No significant difference was noted in terms of presence or absence of symptoms.
(DOCX)

**S3 Fig.  (A) Distribution of routine RT-PCR Ct values across variants.** The significant differences in between the inter-variant median Ct-values were plotted. **(B) Ct values of five primer-probes for each of the 600 samples.** The Ct values of the mutant alleles ranged from 13 to 38, similar to routine RT-PCR Ct values. The variants were named according to the signals obtained from the variant PCR. The control mutation was D614G (not shown in the graph).
(DOCX)

## Acknowledgments

We highly acknowledge our donors for their continuous support and commitment to Institute for developing Science and Health initiatives (ideSHi) and Bangladesh Institute of Tropical and Infectious Disease (BITID). We also acknowledge the initiative taken by the GABRIEL network for this collaborative study. We especially thank the Institute of Epidemiology, Disease Control and Research (IEDCR) for ethical and technical support.

## Author Contributions

**Conceptualization:** Mohabbat Hossain, Mokibul Hassan Afrad, Md. Shakeel Ahmed, Jean-Luc Berland, Florence Komurian-Pradel, Firdausi Qadri.

**Data curation:** Saikt Rahman, Emilie Westeel, Mohammad Tanbir Habib, Manjur Hossain Khan, Jean-Luc Berland.

**Formal analysis:** Zannat Kawser, Saikt Rahman, Emilie Westeel, Mohammad Tanbir Habib, Sharmin Aktar Mukta.

**Funding acquisition:** Jean-Luc Berland, Florence Komurian-Pradel.

**Investigation:** Zannat Kawser, Mohabbat Hossain, Md. Rakibul Hassan Bulbul, Md. Zahirul Islam, Md. Shakeel Ahmed.

**Methodology:** Zannat Kawser, Mohabbat Hossain, Md. Rakibul Hassan Bulbul, Md. Zahirul Islam.

**Project administration:** Zannat Kawser, Jean-Luc Berland, Florence Komurian-Pradel, Firdausi Qadri.

**Resources:** Zannat Kawser, Mokibul Hassan Afrad, Jean-Luc Berland, Florence Komurian-Pradel.

**Software:** Zannat Kawser, Emilie Westeel, Mohammad Tanbir Habib, Sharmin Aktar Mukta, Jean-Luc Berland.

**Supervision:** Zannat Kawser, Md. Zakir Hossain, Mokibul Hassan Afrad, Tahmina Shirin, Md. Shakeel Ahmed, Jean-Luc Berland, Florence Komurian-Pradel, Firdausi Qadri.

**Validation:** Emilie Westeel.

**Visualization:** Mohammad Tanbir Habib.

**Writing – original draft:** Zannat Kawser, Saikt Rahman, Emilie Westeel.

**Writing – review & editing:** Zannat Kawser, Saikt Rahman, Emilie Westeel, Mohammad Tanbir Habib, Mohabbat Hossain, Md. Rakibul Hassan Bulbul, Sharmin Aktar Mukta, Md. Zahirul Islam, Md. Zakir Hossain, Mokibul Hassan Afrad, Manjur Hossain Khan, Tahmina Shirin, Md. Shakeel Ahmed, Jean-Luc Berland, Florence Komurian-Pradel, Firdausi Qadri.

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
