## [Decision Letter · Decision Letter 0]

24 Jun 2024

PONE-D-24-15584SARS-CoV-2 variant survey: comparison of RT-PCR screening with NGS and variant distribution across two divisions of BangladeshPLOS ONE

Dear Dr. Kawser,

Thank you for submitting your manuscript to PLOS ONE. After careful consideration, we feel that it has merit but does not fully meet PLOS ONE’s publication criteria as it currently stands. Therefore, we invite you to submit a revised version of the manuscript that addresses the points raised during the review process.

We look forward to receiving your revised manuscript.

Kind regards,

Haitham Mohamed Amer, PhD

Academic Editor

PLOS ONE

 [This study was funded by Fondation Mérieux.MRHB and ZI are supported by the Fondation Mérieux. ZK and TH were supported by Global Health Fellowship awards, USA. ZK, along with SR, SAM, and MHA, is also supported through programs funded by the National Institutes of Health, including the Fogarty International Center, Training Grant in Vaccine Development and Public Health (TW005572). ].  

Reviewer's Responses to Questions

**Comments to the Author**

1. Is the manuscript technically sound, and do the data support the conclusions?

Reviewer #1: Partly

Reviewer #2: Yes

2. Has the statistical analysis been performed appropriately and rigorously? 

Reviewer #1: N/A

Reviewer #2: Yes

3. Have the authors made all data underlying the findings in their manuscript fully available?

Reviewer #1: No

Reviewer #2: Yes

4. Is the manuscript presented in an intelligible fashion and written in standard English?

Reviewer #1: Yes

Reviewer #2: Yes

5. Review Comments to the Author

Reviewer #1: The authors in the manuscript with number PONE-D-24-15584 studied the variant types of SARS-CoV-2 in 600 SARS-CoV-2-positive samples from two divisions of Bangladesh by variant RT-PCR targeting five mutations of the spike gene (N501Y, P681R, L452R, E484K, E484Q). Among the 600 samples, 99 were sequenced by Nanopore sequencing, the variants from which were compared with variant RT-PCR, indicating 92% consistence. Therefore, the variant RT-PCR is rapid and easily accessible to determine the variants of SARS-CoV-2 in pandemic. There are three main concerns here:

1. In the abstract, the authors mentioned NGS for several times. However, in the section of Materials and Methods, the MinION sequencing via Oxford Nanopore Technology was used in the description of "Targeted next-generation sequencing using the MinION platform". Actually, Nanopore sequencing is a Third-Generation sequencing technique other than NGS. Therefore, it is a basic mistake.

2. The variant RT-PCR was not described in details. What were the sequences of all the sequences of the five sets of the primer/probes? Were those probe/primer sets sensitive to discriminate the mutants? What were the Ct values of the five probe/primer sets? Were the Ct values of the five probe/primer sets close to the Ct of SARS-CoV-2 for the determination? As a method to compare with NGS or TGS, the details should be described well.

3. From December 2021 till now, Omicron has dominated. Many Omicron variants become VOI or VUMs. Can the five mutations represent all the typical variants, especially when Omicron became the dominant variants?

Reviewer #2: The manuscript “SARS-CoV-2 variant survey: comparison of RT-PCR screening with NGS and variant

distribution across two divisions of Bangladesh” is well planned, analyzed, and written study.

Several other studies have demonstrated that variant PCRs can assist genotyping without requiring the resource/infrastructure for WGS. The findings from this study from the geographic region corroborates findings from other studies, though there is limited novelty in the methods.

I have the following minor comments:

104 from eligible NPS with a routine diagnostic RT-PCR Ct value <30 that met the inclusion criteria.

-- Please provide some basic information about the routine assay (assay/kit manufacturer or targets from lab developed tests)

Line 144 : Ct < 27 were selected for sequencing

-- How did you set the thresholds for the selection of samples (30 for variant PCR and 27 for sequencing)?

There are several variant PCRs developed and used over the course to SarsCoV2 evolution, but not much discussed in the introduction or discussion. Please include those to bring up a context during the time of this assay implementation. Like, why did you choose the said assay of the specific mutations and not other mutations used for lineage calls around the time. All of the mutations in the panel seem to be pre-omicron. Please also add in the discussions, how your study compares to similar studies elsewhere. It looks like the results are similar to other similar variant-PCR vs NGS studies, because the different lineages circulated worldwide around similar times, with different time lags according the waves of propagation.

-- can you please add an interpretation table for the various combinations of mutations?

-- you may add the comparison tools from outbreak.info to see if the limited set of 5 mutations were sufficient or deficient in which aspects.

Figure 1 is not necessary. A statement is already enough.

Figure 2 would be better visible with line graph. Please use both color and pattern to make them easily distinguishable.

--Do you think its possible to add the new positive cases from the two centers combined over the time period? That would help visualize the distribution of sampling.

--You can also add a similar plot form GISAID or nextstrain.org for comparison with global data. (or put this as a comparison to figure 3)

-- It would have been easier if all the tables in the multiple supplementary files were placed together in one document.

6. PLOS authors have the option to publish the peer review history of their article (what does this mean?). If published, this will include your full peer review and any attached files.

Reviewer #1: No

Reviewer #2: No

---

## [Author Response · Author response to Decision Letter 0]

5 Sep 2024

We are very thankful to all the respected editors and the reviewers for their valuable insights and comments. We have tried to address all the comments and our responses are given below:

Reviewer #1: The authors in the manuscript with number PONE-D-24-15584 studied the variant types of SARS-CoV-2 in 600 SARS-CoV-2-positive samples from two divisions of Bangladesh by variant RT-PCR targeting five mutations of the spike gene (N501Y, P681R, L452R, E484K, E484Q). Among the 600 samples, 99 were sequenced by Nanopore sequencing, the variants from which were compared with variant RT-PCR, indicating 92% consistence. Therefore, the variant RT-PCR is rapid and easily accessible to determine the variants of SARS-CoV-2 in pandemic. There are three main concerns here:

1. In the abstract, the authors mentioned NGS for several times. However, in the section of Materials and Methods, the MinION sequencing via Oxford Nanopore Technology was used in the description of "Targeted next-generation sequencing using the MinION platform". Actually, Nanopore sequencing is a Third-Generation sequencing technique other than NGS. Therefore, it is a basic mistake.

Response: We agree and to comply with the comment, the term “NGS” was revised to TGS (lines 4,36,48,94,150,189,266,267,272,275,283,299,314,340,345,351,353,355, 366) as suggested.

2. The variant RT-PCR was not described in details. What were the sequences of all the sequences of the five sets of the primer/probes? Were those probe/primer sets sensitive to discriminate the mutants? What were the Ct values of the five probe/primer sets? Were the Ct values of the five probe/primer sets close to the Ct of SARS-CoV-2 for the determination? As a method to compare with NGS or TGS, the details should be described well.

Response: Details have been added to better describe the variant RT-PCR method. The sequences of 5 probes were designed by the manufacturer (Applied BioSystems by Thermo Fisher Scientific), and as we used the commercially available primer-probes and did not design the primers by ourselves, we do not have the sequences. Yet, further communication with the manufacturer can be attempted to obtain the sequences of 5 primer probe sets if required. The primer probes were sensitive to discriminate the mutants within the study time frame, as in our study, Variant RT-PCR successfully distinguished variants of concern (VOCs) and variants of interest (VOIs) in 98.5% of the samples (line no. 40). The Ct values of the five primer probes in the 600 study samples, were congruent with the Ct values of the routine SARS-CoV-2 RT-PCR. The Ct values of these five primer probes for each of the 600 samples have been added as supplementary file 2 and accordingly added in the result section of the manuscript (line 237-238). 

3. From December 2021 till now, Omicron has dominated. Many Omicron variants become VOI or VUMs. Can the five mutations represent all the typical variants, especially when Omicron became the dominant variants?

Response: The five mutations targeted in the study could differentiate the variants based on the signature mutations present in each variant. These five mutations represented the probable variants and we opted to distinguish among major variants, not lineages. For detecting all the lineages of Omicron, which has been dominating since December 2021, inclusion of more primer probes following this variant RT-PCR approach considering the Clade Defining Mutations (CDMs) of the respective lineages of Omicron will be needed. At the time the study was designed, Omicron did not exist. Thus, the five primers/probes' sets can detect but not discriminate sub-variants of Omicron.

Reviewer #2: The manuscript “SARS-CoV-2 variant survey: comparison of RT-PCR screening with NGS and variant distribution across two divisions of Bangladesh” is well planned, analyzed, and written study.

Several other studies have demonstrated that variant PCRs can assist genotyping without requiring the resource/infrastructure for WGS. The findings from this study from the geographic region corroborates findings from other studies, though there is limited novelty in the methods.

I have the following minor comments:

104 from eligible NPS with a routine diagnostic RT-PCR Ct value <30 that met the inclusion criteria.

-- Please provide some basic information about the routine assay (assay/kit manufacturer or targets from lab developed tests)

Response: The following basic information about the routine assay has been included from line 103 – 107. 

Then the samples were tested for routine diagnostic RT-PCR assay using commercially available diagnostic kits targeting ORF1ab and N gene. The NPS specimens with Ct value <30 were randomly selected as per the TaqMan™ SARS-CoV-2 Mutation Panel user guide for undergoing Variant RT-PCR.

Line 144: Ct < 27 were selected for sequencing

-- How did you set the thresholds for the selection of samples (30 for variant PCR and 27 for sequencing)?

Response: The Ct threshold of 30 is given per Thermo' Taqman notice: "RNA extracted from SARS-CoV-2 samples with a Ct value of less or equal to 30". For ensuring enough viral load intending better sequencing coverage, a Ct value of 27 was purposively selected.

There are several variant PCRs developed and used over the course to SarsCoV2 evolution, but not much discussed in the introduction or discussion. Please include those to bring up a context during the time of this assay implementation. Like, why did you choose the said assay of the specific mutations and not other mutations used for lineage calls around the time. All of the mutations in the panel seem to be pre-omicron. Please also add in the discussions, how your study compares to similar studies elsewhere. It looks like the results are similar to other similar variant-PCR vs NGS studies, because the different lineages circulated worldwide around similar times, with different time lags according the waves of propagation.

-can you please add an interpretation table for the various combinations of mutations?

Response: We agree with the comment and accordingly we have added the following in the discussion section (lines 317-320)

“The Thermo Taqman SARS-CoV2 TaqMan SARS-CoV-2 Mutation Panel was one of the first released test. As it is not a kit, one can switch in between primers/probes' sets to adapt to the changing major variants along the time frame. The chosen set was enough to capture the variants along the study time frame.”

The following section has been rewritten (line 336-340) to better explain the reason for selecting the study mutation panel and not other mutations used for lineage calls-

“The variant RT-PCR assay in our study targeted five mutations (N501Y, L452R, E484K, E484Q and P681R) at four positions. This mutation panel was selected based on the signature characteristics of the amino acid substitutions in the spike gene described in NextClade, as these changes are responsible for the emergence of new variants.” As suggested, we have added Table 1 (line 148) for interpreting the various combinations of mutations. 

-you may add the comparison tools from outbreak.info to see if the limited set of 5 mutations were sufficient or deficient in which aspects.

Response: The selected set was sufficient to detect the variants along the study time frame. The sites- https://users.math.msu.edu/users/weig/SARS-CoV-2_Mutation_Tracker.html, https://www.who.int/activities/tracking-SARS-CoV-2-variants, and https://cov-lineages.org/lineage_list.html were studied to compare the globally predominant VOCs and VOIs with the emerging pattern of variants in our study.

-Figure 1 is not necessary. A statement is already enough.

Response: We kept Figure 1 as a supplementary figure, and edited the statement accordingly within the manuscript.

-Figure 2 would be better visible with line graph. Please use both color and pattern to make them easily distinguishable.

Response: We have revised Figure 1 (previous Figure 2) as follows to better understand using color codes for variants.

Fig 1: Distribution and patterns of emergence of the probable SARS-CoV-2 variants detected by RT-PCR over the study time frame

-Do you think its possible to add the new positive cases from the two centers combined over the time period? That would help visualize the distribution of sampling.

We represented both retrospective and prospective cases from the two centers combinedly over the study time frame. Since the study has already been completed, there is no scope to add new cases further from the two centers at present (if we could understand your comment properly). However, we have revised the figure 1 (previous figure 2) distinguishing the retrospective and prospective time frames.

-You can also add a similar plot form GISAID or nextstrain.org for comparison with global data. (or put this as a comparison to figure 3)

Response: We have added another column to the right of the table in the supplementary file 3 for comparison with global data.

- It would have been easier if all the tables in the multiple supplementary files were placed together in one document.

Response: All the supplementary files have been compiled to one document.

---

## [Decision Letter · Decision Letter 1]

29 Sep 2024

SARS-CoV-2 variant survey: comparison of RT-PCR screening with TGS and variant distribution across two divisions of Bangladesh

PONE-D-24-15584R1

Dear Dr. Kawser, 

We’re pleased to inform you that your manuscript has been judged scientifically suitable for publication and will be formally accepted for publication once it meets all outstanding technical requirements.

Kind regards,

Haitham Mohamed Amer, PhD

Academic Editor

PLOS ONE

Reviewer's Responses to Questions

**Comments to the Author**

1. If the authors have adequately addressed your comments raised in a previous round of review and you feel that this manuscript is now acceptable for publication, you may indicate that here to bypass the “Comments to the Author” section, enter your conflict of interest statement in the “Confidential to Editor” section, and submit your "Accept" recommendation.

Reviewer #1: All comments have been addressed

Reviewer #2: All comments have been addressed

2. Is the manuscript technically sound, and do the data support the conclusions?

Reviewer #1: Yes

Reviewer #2: Yes

3. Has the statistical analysis been performed appropriately and rigorously? 

Reviewer #1: N/A

Reviewer #2: Yes

4. Have the authors made all data underlying the findings in their manuscript fully available?

Reviewer #1: Yes

Reviewer #2: Yes

5. Is the manuscript presented in an intelligible fashion and written in standard English?

Reviewer #1: Yes

Reviewer #2: Yes

6. Review Comments to the Author

Reviewer #1: The authors responded all my comments and added necessary data and files in the supplymentary materials.

Reviewer #2: outstanding comment: instead of saying "commercially available diagnostic kits" please provide the kit name and the manufacturer name.

7. PLOS authors have the option to publish the peer review history of their article (what does this mean?). If published, this will include your full peer review and any attached files.

Reviewer #1: No

Reviewer #2: No

---

## [Editor Report · Acceptance letter]

7 Oct 2024

PONE-D-24-15584R1 

PLOS ONE

Dear Dr. Kawser, 

I'm pleased to inform you that your manuscript has been deemed suitable for publication in PLOS ONE. Congratulations! Your manuscript is now being handed over to our production team.

Kind regards, 

on behalf of

Dr. Haitham Mohamed Amer 

Academic Editor

PLOS ONE